# The DLR $CO_2$ -equivalent estimator FlightClim v1.0: an easy-to-use estimation of per flight $CO_2$ and non- $CO_2$ climate effects

Hannes Bruder<sup>1</sup>, Robin N. Thor<sup>2</sup>, Malte Niklaß<sup>1</sup>, Katrin Dahlmann<sup>2</sup>, Roland Eichinger<sup>2</sup>, Florian Linke<sup>1</sup>, Volker Grewe<sup>2, 3</sup>, Simon Unterstrasser<sup>2</sup>, and Sigrun Matthes<sup>2</sup>

**Correspondence:** Hannes Bruder (hannes.bruder@dlr.de)

**Abstract.** As aviation's contribution to anthropogenic climate change is increasing, the sector aims at reducing its climate effect in accordance with international agreements. The strong and variable non- $CO_2$  effects are complex, making reliable climate effect quantification a necessary first step. To support this, we develop the easy-to-use first-order climate effect estimator for single flights  $FlightClim\ v1.0$ . The tool estimates the flight-specific climate effect with a simplified calculation model, without requiring detailed information on exact routing, amount of fuel burn, or weather conditions.

For this purpose, we first analyze a global flight dataset containing detailed trajectories, associated flight emissions, and climate responses. Similar flights are grouped into clusters, and regression formulas are derived to estimate the Average Temperature Response over 100 years (ATR100) for CO<sub>2</sub> and non-CO<sub>2</sub> effects. To prevent abrupt changes at cluster boundaries, we apply linear smoothing as postprocessing. Second, we compare a Multiple and Symbolic Regression approach, which differ in effort and complexity but offer similar estimation quality. The choice of method depends on the specific application. Both methods are designed for climate footprint assessments due to their simplicity though not suitable for policy measures. Emission trading or monitoring and reporting systems instead require detailed weather and route data to incentivize operational non-CO<sub>2</sub> mitigation. Compared to previous studies, our approach covers more aircraft types, including most commercial airliners, and improves precision through smoothed clustering and a dedicated parameterization of aircraft size influence on the contrail effects.

The resulting climate effect functions are embedded into the Excel-based tool  $FlightClim\ v1.0$ , which implements the formulas of the Multiple Regression approach due to slight qualitative advantages. Requiring only aircraft size and origin-destination airports as input, FlightClim estimates climate effect for  $CO_2$ ,  $H_2O$ ,  $NO_x$  emissions and contrail-induced cloudiness. It includes per seat allocation and supports different climate metrics.

#### 20 1 Introduction

Global aviation more than doubled from 2006 to 2019 in terms of revenue passenger kilometers (ICAO, 2015, 2021). The associated  $\rm CO_2$  emissions grew by 40% to 1036  $\rm Tg(\rm CO_2)yr^{-1}$  during this time span (IEA, 2022). After the considerable reduction of air transportation through COVID-19, it reached pre-pandemic levels again by 2023 and now continues to grow

<sup>&</sup>lt;sup>1</sup>Deutsches Zentrum für Luft- und Raumfahrt (DLR), Institut für Luftverkehr, Hamburg, Germany

<sup>&</sup>lt;sup>2</sup>Deutsches Zentrum für Luft- und Raumfahrt (DLR), Institut für Physik der Atmosphäre, Oberpfaffenhofen, Germany

<sup>&</sup>lt;sup>3</sup>Faculty of Aerospace Engineering, Operations and Environment, Delft University of Technology, Delft, The Netherlands

50

(Economics, 2024). Projections show that aviation's share in global  $CO_2$  emissions could rise from currently about 2% to 22% in 2050 (Cames et al., 2015). This amplifies the pressure on the sector for finding solutions to reach the Paris agreement climate goals.

A number of measures are suited to reduce the climate effect of aviation ranging from technological (i.a. Dahlmann et al., 2016b; Silberhorn et al., 2022; Delbecq et al., 2023) and fuel-related solutions (e.g. Teoh et al., 2022; Märkl et al., 2024; Quante et al., 2025) to operational (i.a. Grewe et al., 2014, 2017b; Lührs et al., 2016, 2021; Teoh et al., 2020; Matthes et al., 2021; Yin et al., 2023; Martin Frias et al., 2024; Sausen et al., 2024) and regulatory options (i.a. Scheelhaase et al., 2016; Larsson et al., 2019; Niklaß et al., 2021, 2025). To be overall effective, these measures do not only need to target the reduction of  $CO_2$  emissions, but also the so called non- $CO_2$  effects. Non- $CO_2$  effects were responsible for about two thirds of the total effective radiative forcing (ERF) in 2018 when considering all aviation emissions from 1940 to 2018 (Lee et al., 2021). Especially the effects of persistent contrail cirrus formation and of  $NO_x$  emissions on the ozone concentration increase the total impact of air traffic on the climate.

Therefore the basis for the development of effective mitigation measures, as well as the first step for climate effect compensation programs is a reliable estimate of the total climate effect of a flight, including the non- $CO_2$  effects. However, while the  $CO_2$  climate effect can be estimated easily, as it is independent of emission source, location and time, the effects of non- $CO_2$  emissions are much more complex to determine (Dahlmann et al., 2023). For simplicity, often the global ratio of non- $CO_2$  to  $CO_2$  climate effects is used as a factor for total climate effect estimation, based solely on  $CO_2$  emissions. An example of this simple estimation option for aviation is the Radiative Forcing Index (RFI, IPCC, 1999), which is the ratio of the total radiative forcing to the radiative forcing of  $CO_2$  emissions.

However, Forster et al. (2006) highlighted the limiting shortcomings of the RFI concept, such as a large variation with time for constant emissions, and concluded that RFI is inappropriate for comparing emissions. In addition, the altitude dependency of non-CO<sub>2</sub> effects has to be considered in the estimation method to avoid misguiding incentives (Faber et al., 2008; Scheelhaase et al., 2016; Niklaß et al., 2019). However, this requires detailed information of the flown trajectory, the aircraft and atmospheric conditions to estimate the various climate effects. To query this data is an elaborate process, public accessibility is limited and the data is not available before the flight. Hence, a simplified estimation method that is easy to use for the climate footprint assessment of single flights yet realistically representing non-CO<sub>2</sub> climate effects is needed.

There are a few methods for simplified climate footprint assessment of single flights publicly available. Popular ones are the "ICAO Carbon Emissions Calculator" (ICAO, 2025), the "Flight Emissions Label" of the European Union (EASA, 2025), the "Aviation 1 Master emissions calculator 2023" of the European Environment Agency (EEA, 2023), Google's "Travel Impact Module" (Google, 2025) and the "myclimate flight emission calculator" (Foundation myclimate, 2025). All of them have particular areas of application and strengths, but all of them only take into account CO<sub>2</sub>-emissions, use constant factors to quantify non-CO<sub>2</sub>-effects or are lacking CiC climate effects. A method that overcomes these shortcomings and only relies on mission parameters as distance and geographic flight region has been introduced by Dahlmann et al. (2023). Dahlmann et al. (2023) analyzed the climate effect of the typical long-haul aircraft type Airbus A330-200 for more than 1000 international city pairs using the climate response model AirClim (Grewe and Stenke, 2008; Dahlmann et al., 2016a) and then fitted altitude and

latitude dependent regression formulas to the AirClim results. The regression formulas enable an easy to use estimation of the climate effect of single flights and show a much better estimation quality than a constant factor. While the root mean square error for a constant factor of 3.4 was about 1.18, the one obtained with the regression formulas was about 0.24, with 95% of the estimates lying within a  $\pm 20\%$  range. However, there is no easy-to-use method available that provides a thorough estimate of the non-CO<sub>2</sub> climate effects for individual passengers or organizations, allowing them to assess their footprint pre-flight for travel decisions and post-flight to track their personal climate impact, without requiring detailed information about the actual flown trajectory, the amount of emissions produced and the prevailing weather situation. Such a method would also enables a quick climate effect estimation for large flightplans, supporting scientific research and organizational carbon accounting.

In the present study, we expand the work by Dahlmann et al. (2023) and develop an easy-to-use estimation method for aircraft climate effects, using climate effect regression functions that are valid for all jet passenger aircraft with a seat capacity of over 20. While Dahlmann et al. (2023) only analyzed one aircraft type, we here analyze the climate effect for various commercial aircraft. Instead of using constant emissions over a typical aircraft lifetime of 32 years, we here use the more realistic assumption of increasing emissions over the next 100 years, which influences the weighting of the individual non- $CO_2$  effects according to Megill et al. (2024). We consider the climate effects of aircraft emissions of  $CO_2$ ,  $NO_x$ , and  $H_2O$  as well as contrail-induced cloudiness (CiC), but exclude the effects of aerosol emissions through aerosol–radiation interactions and aerosol–cloud interactions as the understanding and assessment is not yet mature enough to be included here. This easy-to-use method is only based on the aircraft size as well as the distance and latitude of the flight, and the two latter quantities can be easily computed from the airport pair.

The paper is structured as follows. In the first step, we describe the preparation of the regression flight dataset including a distance and latitude dependent clustering (Section 2). Then we apply both Multiple Regression (MR) and Symbolic Regression (SR) to generate specific climate effect regression functions for each cluster (Section 3). Finally, we compare and discuss the resulting formulas for the climate effect of individual flights (Section 3.4). The resulting equations have been implemented into an easy-to-use estimation tool, for which the user manual is available in Section S5 of the Supplementary Material.

We want to stress here that the method presented in this paper, is not intended to assess the effects of neither individual trajectories, weather situations, specific aircraft or different aircraft generations/technologies.

## 2 Preparation of the regression dataset

The regression formulas for the estimation of the climate effect of single flights are based on a dataset consisting of about 57 thousand flight trajectory simulations. These simulations represent global jet-powered civil aviation, covering approximately 30 million flights on 21 thousand routes between 11 thousand city pairs, which accounts for 98% of globally available seat kilometers (ASK). The dataset is derived from a global flightplan of the year 2012, which serves as the base for the creation of flight emission inventories (Section 2.1). The inventories are then used to derive the climate effect per flight, that represents the dependent variable of the regression (Section 2.2). In the final step of the dataset preparation a clustering is derived to group similar flights for the regression analysis (Section 2.3).

100

105

110

120

## 2.1 Global emission inventory

As the basis for the derivation of regression formulas for climate effect estimation, data from the project WeCare (Utilizing WEather information for ClimAte efficient and ecoefficient futuRE aviation, Grewe et al., 2017a) was used, which was an internal project of the German Aerospace Center (Deutsches Zentrum für Luft- und Raumfahrt; DLR). The project addressed both an improvement of the understanding of aviation-influenced atmospheric processes and an assessment of different mitigation options. An essential output of the project was a new set of emission inventories for global aviation (Grewe et al., 2017a). The network of flight trajectories was developed following a four-layer approach implemented in the AIRCAST method (Ghosh et al., 2016). It is starting from an origin–destination passenger demand network that was built up from exogenous socioeconomic scenarios, via the passenger routes network (sequence of flight segments, a passenger actually travelled from origin to destination) to an aircraft movements network, which assigns aircraft seat categories to the resulting flight routes and provides flight frequency information. The final step is a simulation of trajectories based on the aircraft movements obtained from the aircraft movements network layer using DLR's Global Air Traffic Emissions Distribution Laboratory (GRIDLAB; Linke, 2016). Each mission, defined by departure and arrival cities, aircraft type, and load factor, was simulated under typical operational conditions, resulting in a network of flight trajectories. For this purpose, DLR's Trajectory Calculation Module (TCM; Lührs et al., 2014) was used that applies simplified equations of motion known as the Total Energy Model.

Based on the aircraft's engine state determined by parameters such as thrust and fuel flow, the engine emission distribution of  $NO_x$ , CO, and HC species along the trajectory was determined by applying the Boeing Fuel Flow Method 2 (DuBois and Paynter, 2006). The amount of  $CO_2$  and  $H_2O$  emissions was calculated assuming a linear relationship to the fuel burn. The mapping of emission distributions of all flights onto a geographical grid resulted in 3D inventories. In WeCare, using the approach mentioned above, emission inventories and the corresponding climate effect were estimated for the years 2015 to 2050 in 5-year steps. The forecast was based on the dataset from the reference year 2012. Seven different aircraft seat categories (based on the number of seats) were considered in the inventories (20-50 seats; 51-100 seats; 101-151 seats; 152-201 seats; 202-251 seats; 252-301 seats; 302-600 seats). Each seat category was modeled using one representative jet powered aircraft type (plus one backup aircraft type). The representative aircraft type was selected such that it contributes to a significant share of the respective seat category. Respective engine emission characteristics were taken from the Aircraft Engine Emissions Databank of the International Civil Aviation Organization (ICAO, 2023).

## 2.2 Climate effect estimation

In order to obtain the climate effect for each flight corresponding to the flight plan, the climate effect for each single trajectory is estimated with the non-linear climate response model AirClim (Grewe and Stenke, 2008; Dahlmann et al., 2016a) using gridded emission data for each species. Therefore, AirClim combines 3D aircraft emission data with a set of pre-calculated non-linear emission–response relations for a set of atmospheric locations to estimate the temporal development of the global near-surface temperature change. AirClim includes the effects of the climate agents  $CO_2$ ,  $H_2O$ ,  $CH_4$ ,  $O_3$  and primary mode ozone (PMO) (the latter three result from  $NO_x$  emissions), as well as CiC. For deriving the atmospheric responses for  $H_2O$  and  $NO_x$ -induced

changes, 85 steady-state simulations for the year 2000 were performed with the chemistry climate model E39/CA (Stenke et al., 2009), prescribing normalized emissions of NO<sub>x</sub> and H<sub>2</sub>O at various atmospheric regions (Fichter, 2009). For the effect of CiC, we use atmospheric and climate responses considering the local probability of fulfilling the Schmidt-Appleman criterion as well as ice-supersaturated regions, which were obtained from simulations with ECHAM4-CCMod (Burkhardt and Kärcher, 2011). We follow a climatological approach in the estimation of the climate effect, meaning that the calculated values represent a mean over all weather situations averaging over individual spatially and temporally resolved responses.

For analyzing the climate effect, we assume emissions starting in 2012 and a future increase in emissions according to the scenario Fa1 of the Intergovernmental Panel on Climate Change (IPCC, 1992), which is a reference scenario developed by the International Civil Aviation Organization Forecasting and Economic Support Group (ICAO FESG) with mid-range economic growth and technology for both improved fuel efficiency and NO<sub>x</sub> reduction (IPCC, 1999). Historical emissions are neglected. For background concentrations of CO<sub>2</sub> and CH<sub>4</sub>, which influence the climate effect of CO<sub>2</sub> and CH<sub>4</sub> emissions, we assume IPCC scenario RCP4.5 (Meinshausen et al., 2011). A number of different climate metrics can be applied to account for the different components of the aviation climate effect. However, selecting a suitable metric is challenging due to the uncertainties and varying lifetimes of non-CO<sub>2</sub> effects. Megill et al. (2024) recommend using the average temperature response (ATR) or the efficacy-weighted global warming potential (EGWP) with a time horizon over 70 years. For that reason, we quantify the climate effect using ATR100, which is the mean near-surface temperature change over 100 years. For any climate metric, non-CO<sub>2</sub> effects can be expressed as an equivalent amount of CO<sub>2</sub> emissions, so called CO<sub>2</sub>-equivalents (CO<sub>2,e</sub>), that would produce the same effect over a defined time horizon and a given emission scenario.

AirClim does not account for the influence of different aircraft sizes on contrail climate effect. To account for that we use a parametrization derived from Unterstrasser and Görsch (2014) (see Sec. S1 in Supplementary Material). While this parametrization is already included in the ATR100-values used for the Symbolic Regression (see Sec. 3.2), the MR-formulas for CiC have to be scaled afterwards (see Sec. 3.1).

In the data structure for each of the about 57 thousand simulated flight trajectories, characterized by origin and destination airport as well as aircraft size, the resulting amounts of engine emissions were stored together with the ATR100 climate effect per species. This database was then used to derive the climate effect regression functions as well as regression formulas for fuel use and  $\mathrm{NO}_{\mathrm{x}}$  emissions necessary for the MR-approach.

## 2.3 Clustering of flights by relative climate effects


Due to the large variety of importance of the different climate effect components among different flights, it is challenging to find a single set of equations that would reasonably estimate the climate effect under most circumstances. Therefore, in the first step, we apply a K-Means clustering algorithm to separate the flights into several clusters. This clustering is based solely on the share of the six aforementioned components of the climate effect in the total climate effect:

$$\frac{ATR100_{CO_2}}{ATR100_{tot}}, \frac{ATR100_{H_2O}}{ATR100_{tot}}, \frac{ATR100_{CiC}}{ATR100_{tot}}, \frac{ATR100_{O_3}}{ATR100_{tot}}, \frac{ATR100_{PMO}}{ATR100_{tot}}, \text{and } \frac{ATR100_{CH_4}}{ATR100_{tot}}.$$





**Figure 1.** Clustering of flights, as obtained by the K-Means clustering algorithm (a) and as delineated by simple thresholds (b), shown in the mean latitude–distance space. Each color corresponds to one cluster. We name them the short-flight cluster (green), the tropical cluster (orange), and the mid-latitude cluster (blue).

This ensures that flights in a given cluster have similar climate effect characteristics. The clustering is not directly dependent on proxy quantities to the climate effect, such as the amount and location of the emissions. We use an implementation by the free software machine learning library for the Python programming language scikit-learn (Pedregosa et al., 2011) and scale the input quantities to the standard normal distribution before clustering. We find a partition into three clusters to be most useful, as larger numbers of clusters lead to some cluster distinctions lacking a clear physical interpretation. The resulting three clusters occupy distinct areas in the latitude-distance space (Fig. 1a). We therefore name them the short-flight cluster (green), the tropical cluster (orange), and the mid-latitude cluster (blue).

In the second step, simple thresholds are derived which separate the flights into three categories that approximate the found clusters. This is necessary to enable easy categorization of additional flights not contained in this data set. One threshold is a maximum distance for the short-flight cluster, and another threshold is the absolute mean latitude of great circle trajectories confining the tropical cluster. We choose the values for these thresholds in such a way that the amount of wrongly categorized flights is minimized. This leads to a threshold distance of 462.5 km below which flights are categorized as belonging to the short-flight cluster, and a threshold mean latitude of  $\pm 29.7^{\circ}$  within which flights are categorized as belonging to the tropical cluster. All other flights are categorized into the mid-latitude cluster. This approximation wrongly categorizes 16.8% of all flights used for clustering. The resulting simplified clustering is shown in Figure 1b.

The three clusters have distinct characteristics (Fig. 2). The short-flight cluster has a negligible contribution of contrails to the climate effect at an average of 3.5%, and a strong contribution of  $CO_2$  at an average of 57.4% of the total climate effect. Flights in this cluster are often too short to reach the required altitude of at least about 8 km (e.g.; Kärcher, 2018) for contrail formation. The climate effect of the tropical cluster is dominated by contrails (average contribution of 56.6%) because

Figure 2. Number of flights as function of ratio of the individual climate effect components ( $CO_2$ , CiC,  $H_2O$ , and  $NO_x$ ) to the total climate effect for the 3 flight clusters.

strong contrail formation occurs at tropical latitudes. The mid-latitude cluster contains the remaining flights and has large climate effect contributions from  $NO_x$  and  $H_2O$  (average contributions of 49.1% and 6.8%, respectively; see below for further discussion).

For the cluster analyses only flights with seat category 3 to 7 are used. The remaining seat categories 1 and 2 (less than 100 seats) were only added to the dataset later in development. They contribute only 4.2% to global ASK and therefore a minor share of total aviation emissions. Nevertheless the number of flights with seat category 1 and 2 is high. Therefore, an additional cluster for regional jets was used for MR. For SR all flights were clustered in one of the three clusters.

## 3 Derivation of climate effect regression functions



Based on the dataset described in Sec. 2, we derive climate effect regression functions for each emitted species ( $CO_2$ ,  $NO_x$ ,  $H_2O$ , as well as CiC) separately. These formulas use the size of the aircraft and the locations of departure and arrival airports as input to quickly estimate of the climate effect of individual flights. We explore the use of both MR and SR models for easy-to-use climate effect estimation of individual flights. In the two regression analyses the following quantities are used: flight distance along a great circle d [km], mean latitude along the great circle  $\bar{\phi}$  [°], fuel use f [kg],  $NO_x$  emissions e [kg], maximum takeoff mass (MTOW) m [kg], wing span b [m] and ATR100 [mK].

MR is a widely used statistical approach that models the relationship between a dependent variable (e.g., climate effect of  $NO_x$  emissions) and multiple predefined independent variables, called predictors. The functional relationship between the dependent variable and the predictors has to be predefined. Therefore this approach is especially useful when the factors



influencing the climate effect of the single species are well-understood. However, the predefined functional form may fail to capture more complex, non-linear interactions between variables. On the other hand, SR is an advanced technique that searches for the best mathematical expression to describe the data, offering greater flexibility and the potential to uncover hidden relationships. While SR can model highly complex, non-linear interactions, it requires more computational resources and bears the peril of overfitting. By applying both methods, we aim to identify the approach that most effectively models especially non-CO<sub>2</sub> effects, while prioritizing solutions that offer better accuracy and easy interpretability.

For both methods, we derived regression functions that approximate the climate effect for a particular flight as estimated by using the AirClim model. Following Dahlmann et al. (2023), the total climate effect as expressed by ATR100 can be obtained by the sum of the effects from the individual climate agents, where  ${\rm ATR100_{NO_x} = ATR100_{O_3} + ATR100_{PMO} + ATR100_{CH_4}}$  is the combined climate effect of  ${\rm NO_x}$  emissions:

$$ATR100_{tot} = ATR100_{CO_2} + ATR100_{H_2O} + ATR100_{CiC} + ATR100_{NO_x}$$
(1)

## 3.1 Multiple Regression formulas

Multiple Regression is a common method in environmental science, with primary advantages being its simple application and interpretability. The structure of MR functions is predefined as a sum of predictor dependent terms each multiplied by a coefficient. Each coefficient indicates the impact of a specific predictor, allowing to understand the effects of each variable on the climate. It also enables the inclusion of numerous variables and can incorporate interaction terms of multiple predictors. However, MR assumes a predefined mathematical form making knowledge about the interactions necessary, which can be a limitation when relationships are non-linear. The necessary assumption may lead to misspecification of the model if the actual relationships are not well-captured by these forms. Additionally, MR can be vulnerable to multicollinearity (when predictors are highly correlated), which can distort coefficient estimates.

The MR-approach extends the idea of Dahlmann et al. (2023) and leads to the following structure for the derived formulas for all clusters:

ATR100<sub>tot</sub> = 
$$c_{\text{CO}_2} \cdot f + c_{\text{NO}_x}(d, \bar{\phi}) \cdot e + c_{\text{H}_2\text{O}}(d, \bar{\phi}) \cdot f + c_{\text{CiC}}(d, \bar{\phi}) \cdot d \cdot f_{ACsize}(b)$$
, (2)

where  $f_{ACsize}$  is the adaptation factor for the contrail climate effect due to the wing span b (see Eq. S2 in Supplementary Material), and  $c_{\rm CO_2}$ ,  $c_{\rm NO_x}$ ,  $c_{\rm H_2O}$ ,  $c_{\rm CiC}$  are the cluster-dependent climate effect regression functions. Therefore, the climate effect of a species is estimated as a product of the respective climate effect regression function and the relevant reference quantity (f, e,  $d \cdot f_{ACsize}$ ). These MR-formulas are composed of polynomial and arctan functions and are designed to fit the respective partial climate effects  $c_{\rm CO_2}$ =ATR100 $_{\rm CO_2}/f$ ,  $c_{\rm NO_x}$ =ATR100 $_{\rm NO_x}/e$ ,  $c_{\rm H_2O}$ =ATR100 $_{\rm H_2O}/f$ , and  $c_{\rm CiC}$ =ATR100 $_{\rm CiC}/d$ . The climate effect function for CO<sub>2</sub> is fixed at  $c_{\rm CO_2}$ =8.145×10<sup>-11</sup>mK/kg(fuel), because the climate effect of CO<sub>2</sub> is independent of the emission location in AirClim, so that no fit is required. Details on the derivation of the MR-formulas are given in Section S2.2 in the Supplementary Material.

Note that for the derivation of the climate effect regression functions apart from the predictors d and  $\bar{\phi}$  we use the WeCare estimates for the burnt fuel f and emitted NO<sub>x</sub> e, implying that those are also required for the application of these formulas. If




those are not available we provide additional Fuel and  $NO_x$  MR functions, that only use the flight distance d and seat category (S2.1 in Supplementary Material). For the comparison with the SR-approach in Section 3.4 the derived Fuel,  $NO_x$  and climate effect regression functions are considered, combined determining the quality of the climate effect estimation.

## 3.2 Symbolic Regression formulas

The Symbolic Regression method used here, avoids a pre-defined structure of the formula. Instead an evolutionary algorithm provides a best fit and thereby defines the structure of the formulas. This structure can be represented by an expression tree, called gene. Each node in the gene represents a variable, a mathematical operation or a constant. The nodes are merged to a formula by the tree structure (Koza, 1992). The tool we apply, GPTIPS 2 (Searson, 2015), specifically uses Multi-Gene Symbolic Regression, which combines multiple genes with an additional scaling factor per gene  $(b_1, b_2)$  and a bias term  $(b_0)$  to assemble the whole formula (Fig. 3).

**Figure 3.** Structure of a Multi-Gene Symbolic Regression formula consisting of two genes with factors  $(b_1, b_2)$  and a bias term  $(b_0)$ .

The optimization process to find an optimal formula uses an evolutionary algorithm based on a fitness function, in this case the root mean square error for the given dataset. Beneficiary solutions based on a random start population of multiple formulas are evolved over several generations. The evolution-inspired mechanisms forming the final formulas are fitness-based selection, as well as mutation and crossover (Koza, 1992).

For the derivation of regression functions the flight database is split into 80% training and 20% test data. Four different formulas are computed for the climate effect of the climate agents  $CO_2$ ,  $H_2O$ ,  $NO_x$ , and CiC (see Eq. 1). The two main predictors, d and  $\bar{\phi}$  from the first approach are used in the second one as well complemented by m, that replaces the segmentation into seat categories. The flight distance d is meant to cover effects based on the flight length like fuel use,  $\bar{\phi}$  geographically differing climate effects of emissions and m different aircraft sizes. The dependent variable of the SR-formulas is the ATR100 for  $CO_2$ ,  $H_2O$ , CiC and  $NO_x$ .

To check the effectiveness of the clustering derived in Sect. 2.3, regression formulas with and without use of the three clusters are computed. In the clustered version, separate formulas are derived for each cluster. This leads to in total twelve formulas for the clustered version and four for the unclustered one. For each resulting formula of the clustered and unclustered version a multiple runs of GPTIPS 2 (1296 for unclustered and 648 for clustered) are executed as part of a gridsearch for the regression hyperparameters. The main settings of the GPTIPS-software are used as hyperparameters. The reason for the selected gridsearch-approach with many runs is the high variability in the resulting estimation quality of regression formulas.



Figure 4. Pareto-optimal solutions for a Symbolic Regression of the climate effects with respect to  $\mathbb{R}^2$  and number of nodes by using the unclustered data (blue) and a combined pareto-front of the clustered data (purple). The pareto-optimal solutions, that are chosen, are indicated in red and green.

From all derived formulas the Pareto-optimal individuals according to the coefficient of determination  $R^2$  (Eq. 3) and the number of nodes are considered as candidates for the final formula of the species and cluster (see Sec. S3.1 in Supplementary Material). To obtain one formula for the total climate effect the formulas for  $CO_2$ ,  $H_2O$ , CiC and  $NO_x$  have to be combined according to Equation 1. In this step the numbers of nodes for the  $ATR100_{tot}$  add, but the quality of estimation measured as  $R^2$  has to be newly computed. It is not apparent, which Pareto-formula to choose for each species to achieve an optimum in estimation quality and number of nodes for  $ATR100_{tot}$ . However, by trying all combinations it is possible to identify Pareto-optimal combinations that represent a optimal trade-off between a high value of  $R^2$  and a low number of nodes. Figure 4 shows these  $ATR100_{tot}$ -Pareto-fronts for the unclustered (blue) and the aggregated clustered version (purple; for the individual clusters please see the Supplementary Material, Figure S9). The final choices made are indicated by red and green dots. The selected formulas are given in the Supplementary Material in Section S3.1.

$$R^{2} = 1 - \frac{\sum_{i=1}^{N} (ATR100_{\text{tot}}^{\text{pred}} - \overline{ATR100_{\text{tot}}^{\text{act}}})^{2}}{\sum_{i=1}^{N} (ATR100_{\text{tot}}^{\text{act}} - \overline{ATR100_{\text{tot}}^{\text{act}}})^{2}}$$
(3)

For  $ATR100_{tot}$  in the short-flight cluster the clustered approach shows a significantly better estimation quality than the unclustered one (see Supplementary Material, Fig. S10). For the two other clusters the quality is comparable. Therefore as a combination of low complexity and a high quality of estimation the clustered formulas are applied for flights in the short-flight cluster in the further analysis and the unclustered formulas are used for mid-latitude and tropical cluster flights.

**Figure 5.** ATR100 estimates for flights with an Airbus A320 over the cluster boundary distance of 462.5km depending on the mean latitude. The plot shows the estimation with the unclustered SR-formulas, the formulas for the short-flight cluster as well as the smoothed version, taking the mean of the partially largely differing estimates.

## 3.3 Smoothing of regression formulas at the cluster boundaries

The use of different regression formulas for the derived clusters leads to discontinuities at the cluster boundaries that do not reflect real behavior and might result in disincentives. The significant difference in estimated climate effect over the cluster boundaries (e.g. see Fig. 5) makes it necessary to smooth this effect. The smoothing is implemented by using a weighted sum of the cluster-specifically computed climate effects. The weighting factor evolves linearly from a starting point inside the particular cluster until the cluster boundary. At the cluster boundary the weighting of both cluster formulas is equal. Figure 6 sketches this general scheme of the smoothing. The climate effect of a flight within the smoothing area is accordingly estimated by

$$ATR100 = \begin{cases}
ATR100_{C1} \cdot (0.5 + \frac{|d_{C1,2}|}{2 \cdot b_{C1}}) + ATR100_{C2} \cdot (0.5 - \frac{|d_{C1,2}|}{2 \cdot b_{C1}}), & \text{if } d_{C1,2} \in [-b_{C1}, 0[\\
ATR100_{C1} \cdot (0.5 - \frac{|d_{C1,2}|}{2 \cdot b_{C2}}) + ATR100_{C2} \cdot (0.5 + \frac{|d_{C1,2}|}{2 \cdot b_{C2}}), & \text{if } d_{C1,2} \in [0, b_{C2}],
\end{cases} \tag{4}$$

with  $ATR100_{C1}/ATR100_{C2}$  as the cluster 1 / 2 estimates,  $d_{C1,2}$  the distance from the cluster boundary and  $b_{C1}/b_{C2}$  the smoothing boundary 1 / 2 values. The smoothing boundaries mark the starting points of the smoothing area and are derived as the  $R^2$ -optimal values within preset boundaries.

For both approaches smoothing is applied to the existing cluster boundaries of the recommended versions. The details on the individual smoothing are outlined in the Supplementary Material in the Sections S2.3 and S3.2.

#### 3.4 Comparison of climate effect regression approaches

The climate effect functions were developed to represent a fitting of more detailed results from the non-linear response-model AirClim with algebraic relationships. Hence, the reliability of representing the estimated total climate effect is influenced by the applied fitting procedure. To evaluate the quality and reliability of the estimates, the derived, smoothed formulas from MR



**Figure 6.** Concept for smoothing of regression results at cluster boundaries. The smoothing takes place linearly in a predefined range (smoothing boundary) on both sides of the cluster boundary.

and SR are compared in this section. For SR, the formulas estimate the ATR100 directly leading to one formula per cluster and species. For MR the climate effect functions have to be computed for each cluster, which are four formulas per species apart from CO<sub>2</sub>, on the one hand, as well as the regression formulas for the used reference quantity on the other. Those are seven formulas for the fuel regressions, one per seat category, and 14 formulas for the NO<sub>x</sub> regressions, two per seat category. The last formula of the MR-approach is the subsequent contrail wing span adaption. In total, this leads to 35 formulas for MR compared to 8 formulas for SR without smoothing (see Tab. 1).

One advantage of the MR-approach are the separate fuel and  $NO_x$  functions, which the SR-approach does not include directly, hence fuel can still be derived from the  $CO_2$  climate effect. Furthermore all MR-formulas have the same predefined structure, while each SR-formula is different in shape and operators. Also, even though the SR-approach is optimized towards minimum formula complexity, it generally tends to include irrelevant, over-fitting terms and does not include certain input parameters into formulas for species, where correlations are present (e.g.  $\bar{\phi}$  into  $ATR100_{NO_x}$ ). As advantages the SR-approach evolves according the optimum predictive accuracy and yields a better ratio of complexity in terms of the number of formulas to quality. In addition it enables a continuous estimation over the aircraft size by using the MTOW instead of categorical seat categories.

The estimation quality of both approaches is similar (Tab. 1). Overall, the SR-formulas show a slightly higher coefficient of determination  $R^2$  (Eq. 3). This might result from the optimization towards  $R^2$  in the SR-approach, even though for  $CO_2$  and CiC the MR-formulas show a slightly greater  $R^2$ . In terms of the mean absolute relative error (MARE; Eq. 5) the MR-formulas surpass the SR ones. This mainly results from the better relative estimation for short and medium range flights compared to the long range flights (see Fig. 7). The better estimation of longer flights of the SR-formulas is a result of the absolute error-based evolutionary optimization process, which gives a greater weight to longer flights with higher climate effect. The MARE for

Table 1. Comparison of Multiple Regression and Symbolic Regression formulas for the estimation of the climate effect of individual flights. Quality of fit is quantified by  $R^2$  and MARE. Due to zero or almost zero values in the dataset MARE is not defined for  $ATR100_{H_2O}$  and  $ATR100_{CiC}$ . The number of formulas is counted before smoothing. The first value is the number of formulas for the climate effect estimation and the second for supporting equations like the fuel and  $NO_x$ -regressions.

| Regression formulas                |     | $R^2$  | MARE    | number of formulas |
|------------------------------------|-----|--------|---------|--------------------|
| ATR100 <sub>CO2</sub> :            | MR: | 0.9972 | 5.03%   | 1 + 7              |
|                                    | SR: | 0.9940 | 5.86 %  | 2                  |
| ATR100 <sub>H2O</sub> :            | MR: | 0.8613 | -       | 4 + 7              |
|                                    | SR: | 0.9233 | -       | 2                  |
| ATR100 <sub>NO<sub>x</sub></sub> : | MR: | 0.9529 | 13.38 % | 4 + 21             |
|                                    | SR: | 0.9807 | 20.23 % | 2                  |
| ATR100 <sub>CiC</sub> :            | MR: | 0.8960 | -       | 4 + 1              |
|                                    | SR: | 0.8868 | -       | 2                  |
| ATR100 <sub>tot</sub> :            | MR: | 0.9619 | 20.71%  | 35                 |
|                                    | SR: | 0.9684 | 25.82 % | 8                  |

Figure 7. Trend comparison of the MARE for the  $ATR100_{tot}$  estimation with the Multiple and Symbolic Regression approach over the flight distance.

 $_{100}$  H $_{100}$  and CiC cannot be calculated due to flights with almost or exactly zero ATR100 in the dataset distorting the relative metric.

$$MARE = \frac{1}{N} \sum_{i=1}^{N} \left| \frac{ATR100_{\text{tot}}^{\text{act}} - ATR100_{\text{tot}}^{\text{pred}}}{ATR100_{\text{tot}}^{\text{act}}} \right|$$
 (5)

Figure 8. For Multiple Regression, correlation of estimated ATR100 of  $CO_2$ - (a),  $H_2O$ - (b),  $NO_x$ -emissions (e) and produced contrails (d) with the AirClim estimates (referred here as to "actual"), as well as the  $ATR100_{\rm tot}$ -estimation for the dataset (c) and a validation dataset (f). The color code indicates the flown distance.

Figure 9. Same as Figure 8 for Symbolic Regression.

The MARE of the climate effect estimation is generally higher for short flights, as for these the variety in flight trajectories and non-CO<sub>2</sub>-effects generation increases (Fig. 7). The correlation of both approaches with the AirClim estimated values is





shown in Figure 8 (for MR) and 9 (for SR). For the estimation of ATR100<sub>CO2</sub> the MR-formulas show a better performance than those of the SR-approach, especially for long distance flights, because the points in plot 8a are located closer to or almost on the diagonal compared to plot 9a. The SR-formulas generally tend to underestimate those flights with two noticeable groups of flights being overestimated. These two groups are also distinguishable in the MR-plot 8a. One of them is estimated better and the smaller one is instead underestimated.

ATR $100_{\mathrm{H}_2\mathrm{O}}$ -estimation (see plots 8b and 9b) shows relevant differences between both approaches with the MR-formulas generally overestimating the climate effect especially for long flights. The SR-formulas show a better accuracy for those flights and do in general neither tend to over- nor underestimate.

The quality of estimation for the climate effect of contrails is similar for both approaches (see plots 8d and 9d). As the occurrence of contrails is hard to predict and model, the accuracy of the CiC formula is low. The calculation of a meaningful MARE for contrails is not possible for short flights due to some flights with zero or close to zero ATR100 values, but for longer flight distances the MARE is by 2 to 4 times higher than that of  $ATR100_{tot}$ .

For  $ATR100_{NO_x}$  the SR-approach leads to a better quality of estimation, with fewer points far away from the diagonal in plot 9e than for MR in plot 8e, indicating fewer large estimation errors. In contrast to the SR-formulas the MR-formulas have a tendency to under- or overestimate some distinguishable groups of flights.

The  $ATR100_{tot}$  correlations in plots 8c and 9c show only minor differences between the two approaches. Hence we can conclude that the total quality of both approaches is similar, only with certain advantages for single species.

Apart from the results for the used dataset, the performance of the estimated  $ATR100_{tot}$  for a validation dataset is analyzed. The validation dataset includes 439 flights of the German cargo airline EAT. The mainly short and medium haul flights took place with Airbus A300, A330 and Boeing 757 aircraft in 2021 and 2022. The AirClim climate effect estimates based on the real trajectories of these flights serve as the validation reference. The formulas of both approaches show reasonable correlations for the validation dataset, indicating a valid estimation. Longer flights are rather under- than overestimated (see plots 8f and 9f). This trend is stronger for the MR-formulas, which do also have a lower  $R^2$  value for the validation dataset.

Assessing the sum of all ATR100 estimates for the regression dataset shows the SR-approach to be closer to the actual AirClim values than the MR-estimates. The MR-approach estimates are, apart from  $ATR100_{H_2O}$ , in average smaller than the actual values. This results in a lower sum of  $ATR100_{tot}$  and indicates a general trend for underestimation for the MR-approach. The sum of the SR-approach shows neither a tendency to over- nor underestimate.

# 4 FlightClim v1.0 implementation

The derived regression models from MR are implemented in an Excel application called  $FlightClim\ v1.0$ , which is available in the Supplementary Material.  $FlightClim\ offers$  an easy-to-use estimation of  $CO_2$  and non- $CO_2$  climate effects solely based on the aircraft size, as well as origin and destination airports without further knowledge about the actual flight conditions.  $FlightClim\ s$  core is a simple, tabular input mask supporting the estimation of single flights as well as whole flight plans.







Thereby, the tool is suited for individuals estimating the climate effect of a holiday trip, organizations assessing their one year carbon footprint, but also airlines approximating the climate effect of their flight plan.

After a selection of input values in the input mask (climate metric; aircraft size; origin and destination airports; optional: flight frequency and flight class), the interactive tool returns the climate effect of a flight for CO<sub>2</sub>, H<sub>2</sub>O, NO<sub>x</sub> emissions and CiC in the selected metric and as CO<sub>2</sub>-equivalents. If a flight class is entered, *FlightClim* also calculates a statistically backed allocation per passenger (see Sec. S4 in Supplementary Material). In addition to the climate effect, the fuel burn estimate as well as the estimated CO<sub>2</sub> and NO<sub>x</sub>-emissions are returned as intermediate results of the MR-formulas. The interpretation of the inputs is based on two tool-integrated databases for airport coordinates and aircraft characteristics. In the Supplementary Material in Section S5 the user guide of the tool is included.

In *FlightClim* the MR-formulas are implemented. Compared to the SR-formulas they show a slightly better quality of estimation for short- and medium-haul flights, which are dominating long-haul flights in number. In addition the tool's main area of application is seen in Europe, where inner-European short- and medium-haul flights are dominant. The one-time implementation of the MR-formulas makes their greater complexity in terms of number of formulas less relevant. An extended version of *FlightClim* contains the models of both regression approaches and is available upon request, but less suited for ordinary use, due to the necessary choice of model.

## 5 Discussion

The goal of this study is to develop an easy-to-use calculation method for estimating the total climate effect of individual flights, including  $CO_2$  and non- $CO_2$  effects. Two approaches with smoothed formulas from MR and from SR have been compared. Due to the similar estimation quality of both approaches their greatest differences are the number of formulas and the input parameters, which can hence serve as crucial points for making a choice. Therefore, the SR-formulas can be recommended for application, if the complexity of the calculation in terms of the number of formulas is an important factor or if the aircraft size should be modeled continuously. If the estimation quality of short- and medium-haul flights is of greater importance, like for the *FlightClim* implementation, the MR-formulas are the better choice. In general, the specific requirements of an application towards the complexity, interpretability or the quality of estimation should serve as decisive points, which approach to use.

For both approaches applied in this study the ratio between non- $CO_2$  and  $CO_2$  effects is approximately 4 for the used global aviation emission dataset. This number is higher than in other alternative publicly available methods for simplified climate footprint assessment of single flights. They use a constant factor of 2 to 3, which is based on assessments of total historical aviation emissions (e.g., from 1940 to 2018 for Lee et al., 2021). It has to be noted that the relation between non- $CO_2$  and  $CO_2$  strongly depends on the level of the  $CO_2$  reference and the climate metric. Since the regression functions are designed to estimate the climate effect of present and future flights, we do not consider any emissions of historic aviation. Given the long lifetime of  $CO_2$ , historical assessments such as Lee et al. (2021), who analyzed aviation emissions from 1940 to 2018 in terms of ERF, report a stronger dominance of  $CO_2$  (31%) than in the present study (19%). A direct comparison is, however, limited because different metrics (ATR vs. ERF) and emission patterns (historical vs. present and future) are considered. Nevertheless,







the relative importance of non-CO<sub>2</sub> species is broadly similar, with shares of 4 % for  $H_2O$ , 33 % for  $NO_x$  and 44 % for CiC in our dataset versus 2 % for  $H_2O$ , 16 % for  $NO_x$  and 52 % for CiC in Lee et al. (2021), when excluding the studied aerosol effects.

The derived MR-formulas are integrated into the easy-to-use Excel-tool *FlightClim v1.0*. When applying the estimator it is of key importance to consider its limitations. *FlightClim* is based on regression formulas, that themselves fit the results of the climate response model AirClim. This means that the estimation quality and precision is not comparable to complex climate-chemistry models. For example, the developed tool is not suited to compare the climate effect of flights with similar aircraft of different generations or different travel times in the year, meaning that for an individual flight the real climate effect can strongly deviate from the estimated average. It is also not suited to study certain atmospheric characteristics and their impact on the climate effect. To answer those questions more complex models are needed. The main advantage of *FlightClim* is that it produces reasonable estimates including CO<sub>2</sub> and non-CO<sub>2</sub>-effects while being easy to use and requiring very few input data per flight, in fact only origin and destination airport as well as aircraft size.

#### 6 Conclusions

This study presents two methods for an easy-to-use estimate of the climate effect per flight considering  $CO_2$  and non- $CO_2$  effects, of which one is included into the flight climate effect estimator  $FlightClim\ v1.0$ . The tool is made available as an Excel application, which is available in the Supplementary Material. The estimation only depends on the origin and destination airports and the aircraft size (seat category for MR or MTOW for SR). It is independent from information about the actual flights like the flown trajectory, real fuel burn or current weather. Thereby the estimation describes an average in terms of time of the year and day as well as aircraft and assumes great circle trajectories. The estimation methods are based on a global dataset of ATR100 climate effects per flight for  $CO_2$ ,  $H_2O$ ,  $NO_x$  and CiC estimated with AirClim representative for jet aircraft with a capacity of 20 to 600 seats.

Potential use cases for *FlightClim* are advanced analyses on the climate effect of a full year airline, as its effect averages over the year, plausibility checks, or a backup when airlines are unable to provide more detailed data on aircraft and engine used, trajectory and deviations flown, and meteorological conditions on the day of flight. *FlightClim* allows an airline to achieve an initial estimate of the total climate effects of their whole flight network. Additionally it can be used for the extension of online climate effect estimator tools by non-CO<sub>2</sub> effects or to include a comparison of the climate effect of flights into booking platforms considering non-CO<sub>2</sub>-effects. However, when applying the estimator its limitations always have to be considered and the methods must only be used for questions they are able to answer.

Compared to the predecessor study by Dahlmann et al. (2023), we here expand the area of application building on a global dataset representative for a worldwide flightplan and a wide range of jet aircraft instead of only the Airbus A330-200. Moreover, we add a wing span-wise adaption of contrail climate effect to the tool-chain of the regression dataset. To account for the larger scope a clustering is introduced, requiring a smoothing of the estimates at the cluster boundaries. In addition this study does consider two different regression methods and contrasts them. The introduced *FlightClim* tool goes beyond alternative methods



for simplified climate footprint assessment of single flights, because regressions of the climate response include the regional dependency of climate effects, instead of using constant factors for approximating non-CO<sub>2</sub>-effects.

The two utilized methods, MR and SR, differ in effort and capabilities of the methods themselves as well as in quality and quantity of the resulting regression formulas. Even though SR is a more advanced and adaptable method, the estimation quality of the resulting formulas of both approaches is similar. The main advantages of the SR-approach are that it uses the continuous MTOW as aircraft size parameter and is more straightforward and thus less complex. However, the MR-formulas are easier to interpret and yield higher quality results for short and medium range flights. Overall both approaches lead to robust models that enable an easy-to-use climate effect estimation for single flights.

The similar quality of both regression methods indicates, that the resulting estimation quality is not primarily limited by the used method, but rather by the complexity of the database and the regression parameters as well as the settings for the regression analyses. To utilize the whole potential of advanced methods like the symbolic regression those aspects have to be improved first. For example overcoming the limitation of a small number of reference aircraft types included in the dataset could improve the applicability and the overall estimation quality. As another major potential improvement for further studies, the error metric of the regression was identified, as it quantifies the estimation error and serves as the optimization factor during the regression analysis. In this study, error metrics based on the absolute estimation error were used. As the range of values for the climate impact of flights in the dataset is huge due to large differences in aircraft sizes and flight distances, for flights with small climate impacts the relative quality of estimation can drop significantly compared to flights with higher impacts, because of the absolute optimization incentive. Therefore an adjustment in the error metric might be necessary to achieve regressions of a better and more equally distributed estimation quality in further studies and to exploit the whole potential of advanced regression methods.

*Code availability.* The python code used for the clustering and generation of the MR climate effect functions as well as the Matlab code for the derivation of the SR formulas are provided in Bruder et al. (2025, DOI: https://doi.org/10.5281/zenodo.17184041).

Author contributions. R. N. T. performed the clustering, the Multiple Regressions for the climate effect functions, created figures, and wrote a first version of the manuscript. H. B. performed the symbolic Regressions, adopted the manuscript and the figures and created the Excel tool. F. L. simulated the trajectories and created the emissions inventory. K. D. computed the aviation climate effects using AirClim. M. N. created the regressions for fuel use and NO<sub>x</sub> emissions and the Excel tool. S. U. provided the contrail wingspan adaption formulas. All authors helped with discussions, conceptualizing the research and finalizing the paper.

Competing interests. Volker Grewe and Simon Unterstrasser are members of the editorial board of the journal.

Acknowledgements. This work is part of the project "Untersuchung der praktischen Umsetzung der Einbringung von Nicht-CO<sub>2</sub>-Treibhausgas-Effekten im Luftverkehr in das EU-ETS einschließlich Clusteranalyse", funded by the German Environment Agency (Umweltbundesamt – UBA). This work also received funding from CE Delft for regional jets adaptations.

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
