# Peer review of "The DLR $CO_2$ -equivalent estimator FlightClim v1.0: an easy-to-use estimation of per flight $CO_2$ and non- $CO_2$ climate effects"

_EGUsphere, 2025_

## Author Comment (AC1)

Dear Ms. Valcke, dear reviewers,

thank you for your efforts in editing and reviewing our manuscript. We are highly grateful for the detailed reviews received and are happy to hand in the revised version of our manuscript, that profits from the improvements implemented as a result of the raised points by the reviewers.

The full answers to the two reviews are attached below with the original statements of the reviewers in blue and our comments and answers in black, but for clarity we here additionally included brief summaries of the four main adaptions in the revised manuscript.

**1. Scientific significance**

The procedure for the dataset generation is based on atmospheric response surfaces based on chemistry-climate model simulations and a 3-D emission inventory representative for 2012 commercial aviation. AirClim combines both to estimate the per-flight climate effect. FlightClim is a regression model, that estimates that per-flight climate effect based on a reduced number of input variables (see Figure 1 below). Speaking of a mere 'parameterization of a parameterization' is therefore ill-posed. Especially for the estimation of non-CO2 effects by the broad public where simple multipliers are still commonly used, our tool fills a crucial gap for aviation climate effect estimation. Coming from that end of simple multipliers, it is not justified to mark the assumptions made for FlightClim 'oversimplifying'.

[Figure]

Figure 1: High–level modelling workfow FlightClim is based on.

**2. Uncertainties**

Due to the comment of reviewer William Collins we included a new subsection within the discussion on uncertainties and their assessment and included additional paragraphs throughout the manuscript, to emphasize the importance of the topic. However, note that within the scope of this paper, only limited uncertainty analysis is possible. I.e. we develop a quantification of global uncertainties compatible with Lee et al. (2021) leaving differential uncertainties like spatially differing sensitivities or effects of different aircraft technologies for future work.

**3. Comparison and progress over Dahlmann et al. (2023)**

Both reviewers raised the question, how the performance of the model developed compared to its predecessor, the regression model from Dahlmann et al. (2023), that was developed only for the

Airbus A330 aircraft with a spatially focused dataset. In the extensive quantitative comparison now added to the manuscript we show, that the Dahlmann et al. model performs well for Airbus A330 flights, comparable to our models, but is not suited to predict the climate effect of flights of other aircraft categories. Our new models keep a good estimation quality over the full scope of jet aircraft from a small Embraer till a huge A380 and at the same time increase global coverage. To enable this great advance, our approach includes a clustering with additional smoothing and integrates a wingspan dependent contrail climate effect modeling.

**4. Scientific value of comparing two regression models**

The reviewer raises concerns that using two separate regression models is too detailed (although he suggested to study this in his Thor et al. (2023) review) and that there are differences in the design of the two compared regression analyses. However, this level of detail is necessary to study the sensitivity of the regression approach on estimation quality and the differences do not show to have significant influence on the estimation quality, as the estimation quality of both approaches is comparable. From that result we concluded that the errors are not based on too easy regression models, but rather on the dataset, the limited input variables, as well as the regression error metric, and the physical processes themselves. We now made this important result more prominent throughout the manuscript.

Thank you again for considering our submission to the Geoscientific Model Development journal, we look forward to hearing from you soon.

Best regards,

Hannes Bruder, Robin Thor, Malte Niklaß, Katrin Dahlmann, Roland Eichinger, Florian Linke, Volker Grewe, Sigrun Matthes, and Simon Unterstrasser

**Answer to the first Review (RC1) by Professor William Collins:**

**Review of Thor et al. 2025**
Original by the reviewer, correct citation is Bruder et al. 2025.

This paper describes two regression methods for parameterising the outputs of a tool that itself parameterises the CO2 and non-CO2 effects of aircraft flights. It extends a previous parameterisation by allowing for different aircraft types. Given that this study is the parameterisation of a parameterisation is it not obvious that such a detailed regression analysis using two separate methods is warranted.

We want to subdivide the statement into four different parts and discuss the raised points in that order.

1. The first part is, that the reviewer states, that AirClim "parameterises the $CO_2$ and non-$CO_2$ effects of aircraft flights". This is indeed not accurate, as AirClim is not a mere parameterisation, but a climate response model derived from chemistry-climate simulations. It is designed to estimate the non-$CO_2$ and $CO_2$ climate effect of 3-D emission inventories requiring preceding trajectory and emission modeling.

2. We agree to the reviewer´s second point, that the paper extends the previous estimation model by Dahlmann et al. (2023), even though he degrades the change to only allow "for different aircraft types". In fact we also extended the geographic applicability by using a globally representative flights dataset, tested and compared two different regression methods and implemented a clustering, smoothing and a contrail-wingspan adaption. FlightClim does in addition also estimate the absolute climate effect of $CO_2$, $H_2O$, $NO_x$ and contrails, while Dahlmann et al. (2023) only regresses the ratio of a species climate effect to the $CO_2$ climate effect.

3. Related to the first part, we understand the third argument of "Given that this study is the parameterisation of a parameterisation", however, do not fully agree. The first step, or here named first parameterisation, is a general description of the atmospheric response independent of the air traffic and depicts a spatial atmospheric sensitivity to aviation emissions. In our study we combine those climate response surfaces, that go far beyond a parametrisation, with an aviation emission inventory to allow an assessment of individual routes and derive a route and aircraft specific regression model, here named second parameterisation.

   To avoid that this apparent misunderstanding of the reviewer occurs to readers of the paper's final version, we include a high-level overview of the modeling workflow as text and figure into the introduction section. See Figure 1 in the revised version and the following added paragraph:

   *"The method is based on the combination of atmospheric climate response surfaces, that are derived from chemistry-climate model simulations, and a detailed 3-D aviation emission inventory for a full year. From those AirClim estimates the individual flights' climate response that serves as the regression database. The derived FlightClim regression models are route and aircraft specific parameterisations of the flight-specific climate effects. The described modeling workflow is depicted in Figure 1."*

4. As a fourth concern, the author asks if "such a detailed regression analysis [] is warranted."

   We are happy, that the reviewer volunteered again to report on our paper, as he already did for the previous version of this manuscript, Thor et al. (2023). In the review of Thor et al. (2023) (DOI: https://doi.org/10.5194/gmd-2023-126-RC2) the reviewer wrote: "I recommend revisiting this clustering and fitting as it looks by eye as if it could be done a lot better."

As a consequence to this review we withdrew the Thor et al. (2023) manuscript to work on a more advanced modeling approach (the symbolic regression) with a method capable to follow complex dynamics, less dependent on user pre-definitions yet featuring a symbolic form to achieve a better fit. This advanced modeling approach now shows that the errors are not based on too easy regression models, but rather on the dataset and the physical dynamics themselves, hindering a better estimate with a simplified approach based on only OD-pair and aircraft size. The present report of the reviewer raises concerns that using two separate regression models is too detailed. However, this level of detail was obviously necessary to follow the reviewer's recommendation in the first report to study the sensitivity of the regression approach on the estimation quality. However, we thank the reviewer for mentioning this point again, as we missed to make it prominent in the paper, we have now added it in abstract and conclusions.

Also note that in the field of the estimation of non-$CO_2$ effects by the broad public, simple multipliers are still commonly used. Our study shows, that even though only knowing OD-pair of a flight and the aircraft type everyone can estimate a flight´s climate effect much more accurate using FlighClim than with those multipliers (MARE of 21 % instead of 59 %), taking into account geographical and aircraft induced characteristics.

This study needs to account for any uncertainties in the underlying understanding of the non-CO2 climate effects. From Lee et al. 2021 the uncertainties in CiC, H2O, and NOx are approximately 70%, 60% and 100%. It does not seem justified to fit up to 4th order polynomials to such uncertain data. Any errors in the parameterisation will be small compared to the uncertainties in the physics. The function forms of the dependencies on latitude and distance are ultimately derived from a single climate-chemistry model (E39/C) for atmospheric composition appropriate for the year 2000; hence confidence in the precise forms of the dependencies does not seem high enough to justify anything beyond the simplest parameterisations.

We thank the reviewer for the comment. We totally agree that uncertainties in the estimates for the non-$CO_2$ climate effect are large on the global scale. We break the global effect down to individual routes taking into account the spatial atmospheric sensitivity to emissions as well as aircraft characteristics. Therefore the full assessment of uncertainties has to consider both, global uncertainties, e.g. as described by Lee et al. (2021), as well as flight-specific uncertainties, e.g. as a consequence of different spacial sensitivities. Together with the global uncertainties we explicitly discuss those in the newly added Uncertainty-section 5.2. We sketch a methodology to fully assess the uncertainties of the developed tool, hence the estimate of the flight-wise uncertainties is clearly beyond the scope of this paper. But we agree that it must be addressed properly. Therefore we have added a paragraph in the Discussion section and extended the Conclusion section, also stating that a flight-wise uncertainty estimate should be developed and integrated in future versions of FlightClim. For the global uncertainties we developed a quantification inspired by Prather et al. (2025) and in accordance with Lee et al. (2021).

Although the uncertainties of aviation non-$CO_2$ climate effects are large in general, there are for example geographic influences that are well understood and state of the science, such as latitude-dependencies. In addition, flight-physics based processes are represented in the database, which are far more certain than the total climate effects and comprise only small dependencies on the large overall underlying uncertainties. Hence, modeling those non-linear influences does indeed justify the use of higher order models. However, we agree that it is important to always explicitly state, that the higher order of modeled dependencies does not mean, that the precision of the model itself is high. We do have general statements covering that point in the text already, but we included an additional quantification and discussion on the uncertainty of the regression model into the new Uncertainty section, to clarify the reviewer's point. The quantification features distance-dependent confidence intervals complementing the already included regression error metrics like MARE.

The advance over Dahlmann et al. 2023 seems to be the addition of different aircraft types. It needs to be shown to what extent this study reduces the errors compared to Dahlmann et al. and whether this improvement is significant compared to the overall climate uncertainties. Could a simple modification have been made to Dahlmann et al. to account for aircraft size rather than developing two entirely new parameterisations?

The driving advance over Dahlmann et al. (2023) is indeed to extend the scope from just the Airbus A330 aircraft to the whole commercial jet aviation sector from a small Embraer regional jet till the Airbus A380 double-deck long-range aircraft. To enable this massive increase in scope, a lot more than just a new dataset with more aircraft was necessary, that did even though also as a second effect significantly increase global coverage. The inclusion of short range flight raises the issue of large differences in the underlying climate effect characteristics, making a single regression formula for all flights disadvantageous. Therefore a clustering of flights is necessary, which is another major achievement of this study. In the Symbolic Regression approach we tested the unclustered regression and even though Symbolic Regression is highly self-adjusting it was not able to find formulas, that fitted both long and short-range flights well, underlining the necessity of a clustering. Therefore a simple modification of the Dahlmann et al. (2023) formulas, as suggested by the reviewer, does not enable a valid estimation. The steps performed in this study are crucial to the estimation quality of the resulting model we were able to achieve.

We want to thank the reviewer for the suggestion to include a comparison between the model from Dahlmann et al. (2023) and the models developed in this study, even though on a qualitative level the improvement in applicability already stands on its own. We are happy to include the quantitative comparison into the revised manuscript. In a new section in the Supplementary Materials and a summary in the Conclusion section we extensively compare the estimation quality of the Dahlmann et al. model to the MR and SR models based on all Airbus A330 flights of the dataset of this study, as well as the whole dataset with all aircraft. For the A330 flights the models show a comparable quality of estimation, indicating a valid estimate for all of them. But when applying the Dahlmann et al. (2023) formulas to the whole dataset it becomes apparent, that the Dahlmann et al. (2023) formulas are only fitted to the Airbus A330 aircraft and are not applicable to other aircraft categories (R2 of 0.26), whereas the MR and SR models show a comparable performance over the whole dataset (R2 of 0.70 and 0.72), indicating a valid estimation.

Concerning the doubt raised if "developing two entirely new parameterisations" is appropriate, we first want to stress, that the models are far beyond simple parameterisations and add, that the developed Multiple Regression model is also not entirely new, but the logical successor of the regressions of Dahlmann et al. (2023) with the same regression type and similar formulas structure. The experience gained from Dahlmann et al. concerning the regression analysis was fed into the MR approach of this study. The point whether the development of two models is appropriate is already rebut in the answer to the first comment of this reviewer in point 4.

Next to the extension of the scope of applicability we also extended to models´ output compared to Dahlmann et al. (2023). While Dahlmann et al. only estimate the ratio of the climate effect of $H_2O$, $NO_x$ and contrails to the climate effect of $CO_2$, the study provides regression models for the climate effects themselves, also for the climate effect of $CO_2$, increasing the number of potential use cases and making a separate calculation of the $CO_2$ climate effect superfluous.

The meaning of the ATR100 metric needs to be explained. My understanding of the text is that this takes a single flight in 2012 and integrates the temperature out to 2111 and divides by 100. So it is not clear how this relates to an increasing emissions over 100 years or depends on future climate scenarios.

Thank you for your question! We are happy to give a short explanation on the used ATR100 climate metric. To increase clarity about the metric in our manuscript, we extended the respective paragraph in Sec. 2.2.

For analyzing the climate effect, a physical climate metric is selected which assumes growing

emissions for characterizing radiative effect of the single flight emission in a future atmosphere, instead of applying a pulse metric. The reason behind this choice is that there exists a mismatch between the summed effect of pulse emissions and the respective scenario analysis (equal to the effect of aggregated pulse emissions). Therefore, we calculate $CO_2$-equivalents based on a scenario with ATR100 as a metric and apply these equivalents to the pulses in FlightClim. As a consequence, for example when selecting ATR100 as a physical climate metric, the climate effect of these single flight emissions are evaluated assuming a future increase of emissions and concentrations over the next 100 years, which is relevant for the radiative effects estimated as such changing concentrations in the future are taken into account.

Many of the differences between the MR and SR approaches seem to be due to choices made, not due to any inherent features of the approaches e.g. whether to calculate ATR100 per kg fuel or per flight. It is not explained why MR and SR treat aircraft size differently (e.g. use of MTOW vs seat category). Is this because the understanding developed during the course of the study? Or because the regional data became available too late to be included in some of the analysis?

Indeed, there are differences in the setup of the MR- and SR-studies. The MR-approach with seat category originated from Thor et al. (2023). When additionally developing the SR-approach for this study, we opted to reduce number of formulas and complexity as the SR-formulas can include some of the underlying processes in the formulas implicitly. In the MR-approach, in contrast, the processes are separated, which is partly due to the more restricted regression approach. In the course of this development, it was also that MTOW was used instead of seat category to support a finer modelling of the aircraft sizes and to reduce steps in the results. However, despite all these differences in design choices, the estimation quality of both approaches is similar, indicating that these choices are actually not decisive after all. To meet the reviewer's point, we will make the differences in choices for the two approaches more clear in the paper and explain that this does not have a large impact on results more clearly.

It should be made clear who the intended audience for this paper is, particularly to what extent a reader could apply any of the conclusions more generally beyond the specifics of this particular tool.

We thank the reviewer for this suggestion and will happily include such a statement in the paper. The audience is on the one hand the scientific aviation climate effect community as the tool enables quick checks and high-level studies. On the other hand, the paper is directed to developers of first-order aviation climate effect tools e.g. for websites and thereby indirectly also the broader public, as our models enable passengers and organisations to track their climate footprint in a more precise way than currently available tools do (especially referring to simple multipliers that are still used by some institutions), due to the inclusion of non-$CO_2$ effects.

The abstract already includes that the tool is *'designed for climate footprint assessments'*, but we will now additionally include the following statement in the discussion section:

*"FlightClim is suited for climate footprint assessment by passengers to compare individual flights and even different travel modes. Organizations can use it for climate effect monitoring or travel planing. But the tool can also be used for scientific research for a quick quantification of the climate effect of large flight plans, where the restrictions of the developed method are acceptable."*

Specific points

Line 71: The implications of the increasing emissions needs to be explained since ATR100 seems to be for each flight in 2012.

The climate metric is used to derive ratios of non-$CO_2$ to $CO_2$ effects. Unlike many other longer-lived (decades to centuries) non-$CO_2$ species those from aviation that are very short-lived

(hours to weeks) are very sensitive to the emission profile used in the calculation of the climate metric and by this it largely affects this ratio (Dahlmann et al., 2025, see Fig. 4). We argue, in agreement with Megill et al. (2024), that this ratio of non-$CO_2$ to $CO_2$ effects should be consistent with a scenario analysis implying that the sum of pulse emissions that defines a increasing aviation scenario has the same ratio as the sum of the climate effects of the pulses. Therefore we use in the calculation of the climate metric an aviation scenario, derive the climate metric values with specific non-$CO_2$ to $CO_2$ ratios and apply those to a pulse emission of the regarded climatological flight.

Line 125: The use of the year 2000 atmospheric composition (particularly NOx) may bias the NOx effect since this will have changed significantly over the last 25 years. Skowron et al. 2021 showed that this can even change the sign of the NOx effect.

We fully agree with the reviewer that both future changes in the atmospheric composition and the amount of aviation $NO_x$ non-linearly affect the ozone and methane response. That has been shown earlier (Lee et al., 2010, their Table 6) and in principle is included in the underlying chemistry-climate modelling, however, this non-linearity is not included in the response surfaces of AirClim. The AirClim data show indeed negative $NO_x$-RF for low latitude short-range flight, indicating the different nature in the ozone and methane responses (Dahlmann et al., 2023).

Line 132: Why is a future increase in emissions needed if the ATR100 is calculated from flights in 2012? Why is a 1992 scenario used, presumably understanding of future flight patterns was pretty basic 33 years ago?

The increasing emissions are used to better represent actual ratios between $CO_2$ and non-$CO_2$ effects (see answer above). We agree that the used emission development is indeed quite old. However, the decisive factor here is not the exact course of emissions over time, but rather whether emissions increase, remain constant or decrease, as this influences the relative contributions of the short-lived non-$CO_2$ effects and the long-lived $CO_2$ effects. For decreasing emissions, as in the case of pulse emissions, the effect of $CO_2$ emissions is weighted stronger, as $CO_2$ still have an effect while the non-$CO_2$ effects have already subsided. If, on the other hand, rising emissions are taken into account, the non-$CO_2$ effects are weighted stronger, as the cumulative effect of $CO_2$ is not sufficiently taken into account. Therefore, the exact choice of emission development has only a minor effect. As the data was already available from an earlier project in which this emissions development was prescribed, the climate impact was not recalculated using AirClim.

For analysing the effect of different emission developments we included conversion factors in the FlightClim tool to convert the climate metric e.g. to a pulse emission scenario, which is recommended for a single flight perspective and thereby default in FlightClim.

Line 135: The key atmospheric component will be the NOx rather than CO2 and CH4. As above, Skowron 2021 showed that the NOx effect can change sign depending on the NOX background. It is not clear why a future scenario is needed if the flights are for 2012. Does the ATR100 incorporate a varying CO2 radiative efficiency with time? This would be a very different philosophy to all other climate metrics e.g. GWP100 which assume a fixed radiative efficiency over the 100 year time horizon. It may be a valid criticism that metrics should actually be scenario dependent, but given this is different from current practice this argument needs to be made and explained.

Please see the comments above referring to line 71. The use of the ATR100 metric with an increasing scenario is based on the recommendations from Megill et al. (2024) and aim at providing non-$CO_2$ to $CO_2$ ratios that can be applied for pulse emissions of a climatological flight and are consistent with a scenario rather than describing a future atmosphere.

Line 140: Why is the ATR100 used if the recommended metric is ATR70? The ATR metric needs to be explained better here. My understanding is that it is a pulse metric, i.e. the integrated

climate effect of a "pulse" emission where the "pulse" can be 1 kg of fuel burned, 1 km flown or an individual flight. If so, then no future scenario is needed if the flights are assumed to occur in 2012.

We indeed phrased the statement about the recommendation of ATR70 by Megill et al. (2024) ambiguous. It is actually the case that Megill et al. recommend the application of a time horizon of minimum 70 years. As a time horizon of 100 years was used for the Kyoto Protocol and other political applications, we have also opted for 100 years here. We change that in the statement of the manuscript.

Line 155: It is not obvious why these six variables should be useful to cluster the flights (presumably the PMO is simply proportional to CH4 and doesn't add extra information). It seems that the regressions use total NOx rather than split into O3, CH4 and PMO. The aim isn't to find clusters with different strengths of (e.g.) NOx, but to find clusters where the behaviour of the (e.g.) NOx effect is functionally different.

Thank you to the reviewer for the detailed reflection of our manuscript. The clustering focuses on representing the climate effect characteristics and thereby the available climate effect components present are used. The point on PMO and $CH_4$ is correct. Due to the reviewers remark we tested the consequence of including a combination of $CH_4$ and PMO as a single feature instead of two separate. It showed that the cluster boundary between short-flight and tropical/mid-latitude clusters slightly shifts from 462.5 km to 476.3 km, leading to a different classification for 260 flights, which is 0.45 % of the whole flights database. The latitude boundary would only shift by 0.04°, resulting in the same value after rounding. As the test has only shown slight shifts in cluster boundaries, the change in clustering features will be included in further updates of FlightClim, but not changed for this study due to the immense effort required.

The latitudinal clustering appears not to be useful for SR, and in MR causes issues at the latitudinal boundary.

Based on the recommended combination of short-flight and unclustered formulas for the SR-approach and general issues at the cluster boundaries we however solved through the implementation of a smoothing, the question of the usefulness of the latitudinal clustering can arise. As noticed correctly, the latitudinal boundary is the less important boundary compared to the distance-wise one. However, the latitudinal boundary still represents clear physical characteristics, especially latitude dependencies of the CiC and $NO_x$ climate effect and is far more relevant than a potential fourth cluster. That is why we decided to prescribe an amount of three clusters to the K-Means algorithm. For the MR approach the separation into clusters of distinct characteristics helps, because the dynamics these formulas of prescribed form are able to describe are limited. And even in the SR case our results show an increased accuracy when using the clustered version with all three clusters. This version is also the recommended SR one, when accuracy of the result is of primary importance. As described our symbolic regression analysis showed a weakness of the used absolute error-based optimisation metric, which led to the unclustered formulas being insufficient for especially short flights. Therefore, when using the formulas of the SR-approach in every case the short-flight cluster formulas should be used. For longer flights both the clustered formulas as well as the unclustered ones follow the reference and only slightly differ in accuracy. So, the user can choose weather a low complexity or a better accuracy is of greater importance and based on this priority decide to use the unclustered version plus the short- flight cluster formulas or the full clustered version.

Line 265: Since the unclustered results include the short flights which have very different characteristics to the other flights it seems strange to include them in the parameterisations of mid-latitude and tropical. Would it not be better to exclude the short flights to derive the parameterisations for these mid-latitude + tropical flights?

The working hypothesis of our clustering approach was to provide regression formulas that estimate the climate impact of individual flights significantly better than an unclustered approach. We were able to demonstrate this clearly for the short-flight cluster. For the two remaining clusters (mid-latitude and tropic), the differences in performance are smaller, yet clearly visible in Figure S12 in the Supplementary materials. The large difference for short flights is mainly due to the absolute error metric used in our study combined with a distribution of climate effect ATR100 values over multiple orders of magnitude. The selection of regression formulas to be implemented in FlightClim represents a trade-off between complexity and accuracy: the goal is to use a minimal set of formulas while still achieving sufficiently good estimation quality. For this reason, we recommended applying the short-flight cluster and the unclustered version, rather than splitting it into two clusters. Nevertheless, if maximum accuracy is required, the clustered approach should be used (see also answer to previous point).

Line 285: This paragraph needs to be explained better. It seems it was the authors' choice to explicitly separate out the fuel use and NOx emissions in the MR scheme but not in SR. Therefore MR could equally have been designed to directly estimate ATR100 if the authors had chosen to. Similarly use of discrete seat categories in MR rather than MTOW in SR is purely the authors' choice and is not inherent in the types of regression.

Thanks to the reviewer for the suggestion to include an extended explanation, that we think can help the reader to better understand our study design. Therefore we added the following statement:

"The number of formulas and the input parameters are a result of the different study designs for the two approaches and not inherent to the methods, even though the SR approach supports a lower number of formulas by being able to adapt to complex dynamics independently, whereas the MR-approach depends on a predefined structure. Therefore the MR-approach features separate seat categories and a separate estimation of fuel consumption and $NO_x$-emissions to distinguish their dynamics."

And it is correct, that the mentioned differences are partially caused by developing understanding over the study and a shift in design principles for the tool. For future more extended studies it will be interesting to see the separate effect of the single design choices, but we were able to show in this study, that even though applying some modifications the performance of both approaches remains comparable. The influence of those modifications is therefore limited, which is a valuable result and a good starting point for further detailed studies.

Line 292: The authors have chosen not to directly use the fuel use in the SR approach. They could have done so if this was useful.

We appreciate the reviewers comment, and are happy to clarify our motivations of the mentioned design change in the regression analysis. Multiple regression is limited in the dynamics it can model and depends on a prescribed form. Therefore, an intermediate step by calculating the fuel burn helps to split up the total dynamics. For symbolic regression we wanted to test, if the advanced regression method is capable of including the intermediate step by itself through its flexibility in structure and find acceptable estimations without the intermediate step. As a result, it was not possible to find estimates of a better accuracy than the MR approach, but about the same, indicating, that it was able to model the intermediate step by itself.

Line 297: The authors have chosen to use seat categories in the MR approach, presumably they could just as easily have used MTOW as a variable if they wanted a continuous function.

Correct, this is mostly caused by developing understanding over the study and a shift in design principles for the tool. We therefore only considered the MTOW as a potential feature for the SR formulas, which we developed after the withdrawal of Thor et al. (2023). Now the reader has

the option to choose the formulas of the SR approach if a continuous modeling of aircraft size is required. If that is not the case the use of MR formulas is appropriate as well like for our FlightClim tool. Both approaches lead to a similar estimation quality.

Line 302: The authors chose to use the absolute error to optimise the SR; they could have chosen the relative error if they wanted to give a better relative estimate for short and medium flights. It might help the reader if the authors explained that the MR are optimised on a per kg fuel or kg NOx basis, whereas the SR are optimised on a per flight basis. Again this is a choice made by the authors and not inherent in the methodologies.

Thanks to the reviewer for this comment. The use of a relative error was indeed tested, but does lead to dissatisfactory results, due to flights with zero or almost zero contrail and $H_2O$ climate effect. The improvement of the optimisation metric is an open improvement, that is part of our further research. On the second point, please see the above answers.

Line 334: It seems worrying that there is a systematic bias to underestimate ATR100 in the MR approach.

Thanks to the reviewer for raising this concern. Transparently we investigated and included potential biases of the models. As the reviewer mentions an under- or overestimation is in general undesirable, therefore we performed in-depth research on the mentioned overall underestimation of the climate effect in the MR-approach. But it is not es easy as saying the approach is under-estimating and thereby systematically biased, because the mentioned parameter according to that the approach is generally underestimating is the sum of ATR100 estimates of all flights. When summing over the whole dataset, the estimated total climate effect is about 5.4 % smaller than the actual one from AirClim. But when checking how many flights are over- and how many are underestimated, it turns out, that 51 % of all flights are overestimated. The mean signed relative error is -4.3 % indicating that in mean a flight is overestimated by 4.3 %. Therefore, it is hard to make a clear statement about over- or underestimation and we decided to include the information of a sum over the whole dataset into the manuscript. We now added the information, that a mean flight is overestimated by 4.3 %, to clarify, that a general statement is hard to make. The reason for the different results for different parameters is, that flights with larger or smaller climate effect values (e.g. longer and shorter flights) show different mean characteristics. And every parameter reflects different characteristics, putting focus on different groups of flights. For the total ATR100 it is visible in figure 8c, that especially between 2e-5 mK and 5e-5 mK flights are rather underestimated, leading to the lower total sum, while flights of smaller climate effect are rather overestimated (unfortunately barely visible in the figure, due to small absolute values), resulting in a mean relative overestimation. In general, we have the goal to design models of steady estimation quality over the whole area of application and without areas of over- or underestimation, but in practice for such reduced models the means to influence such behavior are limited. In future studies we will try to further improve this point especially by means of working on the optimization metric and using more complex models with extended adaption options.

Line 357: The discussion need to include consideration of the physical uncertainties in the non-CO2 effects. From Lee et al. 2021 it would seem that these are far higher than the parameterisation errors discussed in this study.

Indeed the uncertainties in the non-$CO_2$ effects are important and should be explicitly discussed also in combination with the regression error. Therefore we added paragraphs in the Introduction and Conclusion section and dedicated a full new section in the Discussion to the topic of uncertainties (see corresponding answer above)

Line 405: A simple way of adding the wingspan to the Dahlmann formulae would have been to

multiply eqn 12 of Dahlmann et al. with eqn S2 from this study. This would have been a much simpler procedure than developing two new regression techniques here, and any parameterisation errors would have been likely much smaller than the physical uncertainties.

Applying the CiC wingspan correction from our paper to the regressions of Dahlmann et al. (2023) makes sense for adjusting the calculated CiC climate effect only for the reference A330 aircraft. The reviewer misses the point that we here develop formulae to estimate the effect for various aircraft types. In particular for regional jets and short and medium range aircraft the recommended approach would lead to large inconsistencies.

Corrections

Line 55, "CiC" needs to be defined.

Line 62: Units for 1.18 need to be given.

Thank you very much, the reviewers corrections are implemented now.

**Answer to the second Review (RC2):**

In this study, the authors build the simplified model FlightClim to calculate the climate impact of a flight from the knowledge of aircraft size and origin and destination airports. They build on the work described by Dahlmann et al. (2023) by adding more flights and city pairs to the database used to derive the simplified equations are derived, adding more aircraft types, and clustering flights to help with the regression.

Obviously, we are in the realm of very simplified modelling here. FlightClim is built from many oversimplifying assumptions and from a series of regressions of noisy relationships, so the accuracy of the calculated climate impacts is expected to be very low, as stated by the authors. Still, the proposed tool represents an advance, at least from an atmospheric physics point of view, over the simplest solution of using a global multiplier of CO2 emissions to approximate the climate impact of non-CO2 effects. On that basis, I recommend publication, but it would be important to make a better case for the fitness for purpose of the tool.

Speaking of "very simplified modelling" as well as "many oversimplifying assumptions" is not justified when considering the field the developed model is based in. Especially for the estimation of non-$CO_2$ climate effects by the broad public it is as mentioned still common to use simple multipliers or even only a qualitative statement but no quantification. Our tool goes far beyond such truly oversimplifying approaches and generates an estimate as close to the current scientific understanding as possible from the limited data on a flight publicly accessible. We model latitude, flight distance and aircraft-wise dependencies that are even going as far as including a recently developed parametrisation of contrail climate effect dependency on wing span. Thereby, we make this scientific progress in the field of aviation climate effect accessible to everyone.

The authors suggest that the tool could be of use to the general public that want to estimate the climate impact of their flight, or have it estimated in flight booking platforms, and to compagnies that would like to estimate the total climate impact of their aircraft fleet or business travel. It is true that such use does not require very accurate per-flight estimates, but there are still a couple of important requirements that would need to be demonstrated:

- The first requirement is for the estimation of total climate impact to be reasonably accurate (i.e. that errors compensate across all flights). Lines 60-62 suggest that this aspect was quantified by Dahlmann et al. (2023), so can that comparison be done again here?

Thank you very much for this suggestion. We are happy to include a quantification of the estimation accuracy like present in Dahlmann et al. (2023) and even beyond into the revised manuscript. We implement this in two parts.

As the first part we develop a detailed comparison of the quality of estimation between Dahlmann et al. (2023) and the models of this study featuring not only the MARE and R2 metric used in the reviewed manuscript, but also the root mean squared error (RMSE), as done in the section in Dahlmann et al. (2023) referred by the reviewer. For this comparison we created a new section in the Supplementary Materials, as well as the following new summarising paragraph in the conclusion section:

*"Compared to the predecessor study by Dahlmann et al. (2023), we here expand the area of application building on a global dataset representative for a worldwide flightplan and a wide range of jet aircraft instead of a spatially focused dataset with Airbus A330-200 flight only. We estimate the absolute ATR100 of all species including $CO_2$ instead of just the species´ ratio to the ATR100 of $CO_2$. Moreover, we add a wing span-wise adaption of contrail climate effect to the tool-chain of the regression dataset. To account for the larger scope a clustering is introduced, requiring a smoothing of the estimates at the cluster boundaries. In addition this study does consider two different regression methods and contrasts them. Applying the regression model from Dahlmann et al. (2023) to the dataset of this study for all flights with an Airbus A330-200 aircraft shows, that*

*the MR and SR models even surpass the Dahlmann et al. (2023) formulas in estimating the contrail and total climate effect, but are beaten in terms of $NO_x$ and $H_2O$ climate effect. When looking at the whole dataset of different aircraft the new models outplay the formulas only developed for A330 flights for all species resulting in a MARE of 21.8 % (MR) and 25.9 % (SR) in terms of the ratio of total climate effect to $CO_2$ climate effect compared to 31.5 % for the formulas from Dahlmann et al. (2023). A comparison in terms of $R^2$ illustrates this difference even clearer, where the $R^2$ of 0.27 indicates only a really rough correlation for the Dahlmann et al. (2023) model, wheres MR and SR provide valid estimates (0.70, 0.72). This highlights the advantage of the new models to cover a wide range of civil aircraft. A detailed comparison with Dahlmann et al. (2023) is available in Section S6 of the Supplementary Materials."*

As the second part of a advanced quantification of the estimation quality in the revised manuscript we further detailed on the regression error in the context of a newly included uncertainty assessment. Here we develop distance dependent confidence intervals for the climate effect estimates of FlightClim and discuss their implications. In addition we also take a look at the uncertainties included in the regression dataset, which are also of great relevance to the overall estimation quality and should be considered when evaluating the suitability of the tool for a specific application.

- The second requirement is that comparing two flights should give a reasonable change of identifying the flight with the largest impact. That seems limited to comparing flights between clusters but would not work within the same cluster. Is that correct? Is that good enough?

FlightClim is designed to allow users to compare the climate effect of different flights or even the effect of a flight to other transport modes (e.g. car, train; separate tool for those necessary) when planning a trip. The comparison between flights with the goal to identify the one with the lowest climate effect is meaningfully possible with FlightClim without limitation to within or between clusters. FlightClim models the influence of flight distance, latitudinal location and aircraft size allowing to indicate differences in climate effect resulting from these parameters. Influences of different aircraft within the same aircraft size category (only for MR), of aircraft or aircraft model age, of local weather and of actual flight conditions are not represented in the FlightClim estimate. Different clusters (short-range, mid-latitude, tropic) and aircraft seat categories are primarily included to increase the accuracy of the regression modelling, with the most appropriate cluster assigned automatically to each route.

It seems that the accuracy of FlightClim could be easily improved by using seasonal regressions and taking into account the daytime/nighttime nature of the flight, which is important for contrail climate impact. If I understand well, AirClim is able to provide that information. Why not build that in? That would not make the use of FlightClim more complicated.

Yes, that would be an interesting question. However, AirClim is not able to distinguish between flights at different times of the day or of the year. In AirClim, the climate impact is calculated as if the flight were flown repeatedly throughout the year and an average value is calculated from this. Apart from that, this would make the calculation much more complex and require a large number of additional regression formulae. In addition to the differences between day and night, the weather situation would also have to be taken into account, which would also lead to significantly more inputs being required and would run counter to the aim of this tool.

I am surprised by the very large contribution of NOx to total ATR100 for midlatitude flights in Figure 2. Given the current distribution of air traffic, a large fraction of total flights will be midlatitude flights. It should follow that NOx should represent a large fraction of total non-CO2 effects but that would not be consistent with the assessment of Lee et al. (2021). Is there a way to reconcile those results?

In fact, there is a large number of flights in the mid-latitudes where the $NO_x$ effect accounts for

almost half of the climate effect. However, these are often relatively short-distance flights where only a few contrails form. As a result, the contribution of the $NO_x$ effect to the overall climate effect is not as great as it seems to be in Fig. 2. Figs. 8 and 9 show that the climate effects of CiC and $NO_x$ are of the same order of magnitude, with CiC having a larger effect. This also reflects well the results from Dahlmann et al. (2023).

But when comparing the used AirClim climate response model to comparable models, as Lee et al. (2021) did, one can notice, that in average AirClim estimates higher values for the $NO_x$ climate effect than the most other tools (sees new Section S5 in Supplementary Materials). The main reason for this are differences in $O_3$ and PMO estimates.

Other comments:

• Lines 131-134: That scenario is ancient! Given that the assumption of increasing emissions is given as a strength of the new method (line 71), it is surprising that the selection of scenario does not warrant more care. Why not base the calculations on recent aircraft manufacturer forecasts, or on the baseline LTAG assumptions?

Please note that a scenario is defined by a combination of different parameters. First, the reference year determines the modeled flight plan, in particular how often each aircraft type was operated on individual routes. An update is meaningful if the route structure or aircraft allocation has changed significantly over the year; we plan to update the flight plan database for the next FlightClim release. When selecting the climate metric, we also specify the indicator (e.g., ATR, EGWP), the time horizon (20, 50, 70, 100 years), and the emissions scenario per route (constant or increasing emissions). For reasons of comparability, we do not vary the emissions scenarios across routes but apply standardized assumptions.

However, the decisive factor here is not the exact course of emissions over time, but rather whether emissions increase, remain constant or decrease, as this influences the relative contributions of the short-lived non-$CO_2$ effects and the long-lived $CO_2$ effects. For falling emissions, as in the case of pulse emissions, the effect of $CO_2$ emissions is weighted stronger, as $CO_2$ still have an effect while the non-$CO_2$ effects have already subsided. If, on the other hand, rising emissions are taken into account, the non-$CO_2$ effects are weighted stronger, as the cumulative effect of $CO_2$ is not sufficiently taken into account. Therefore, the exact choice of emissions trajectory has only a minor effect. As the data was already available from an earlier project in which this emissions trajectory was prescribed, the climate impact was not recalculated using AirClim. For analysing the effect of different emission developments we included conversion factors in the FlightClim tool to convert the climate metric e.g. to a pulse emission scenario, which is recommended for a single flight perspective and thereby default in FlightClim.

• Line 139: "for that reason" – Megill et al. 2024 recommends ATR70 yet ATR100 is chosen here, so the justification does not quite work.

Thanks to the reviewer for the careful review, we indeed phrased the statement about the recommendation of ATR70 by Megill et al. (2024) ambiguous. It is actually the case that Megill et al. recommend the application of a time horizon of minimum 70 years. As a time horizon of 100 years was used for the Kyoto Protocol and other political applications, we have also opted for 100 years here. We change that in the statement of the manuscript.

• Line 269: "disincentives" for kind of decisions? As stated elsewhere, the tool should only be used for trivial decision making.

Thank you, we changed 'disincentives' to 'inconsistencies'.

• Line 267, Section 3.3: What is the reason for the choice of two different smoothing bound-

The distance boundary of the SR-approach is the reason for this choice. It is beneficial in terms of overall estimation quality to have a larger smoothing boundary for distances longer than the cluster boundary, than for flight distances smaller than the cluster boundary.

• Lines 379-380: "FlightClim is based on regression formulas, that themselves fit the results of the climate response model AirClim." – you are missing another clause: ", which itself fit the results of climate model simulations".

We did now include this useful note, thank you very much to the reviewer.

• Lines 381-384: Must repeat here that the tool is not suited for any kind of trajectory optimisation to reduce the climate impact of a flight.

Thanks to the reviewer for this comment. We did include an explicit statement that trajectory optimization is not a use case of FlightClim.

**References**

Dahlmann, K., Grewe, V., Matthes, S., and Yamashita, H.: Climate assessment of single flights: Deduction of route specific equivalent $CO_2$ emissions, International Journal of Sustainable Transportation, 17, 29–40, https://doi.org/10.1080/15568318.2021.1979136, 2023.

Dahlmann, K., Matthes, S., and Grewe, V.: Conversion of climate metrics for policy applications, https://doi.org/10.5281/zenodo.16355781, 2025.

Lee, D., Pitari, G., Grewe, V., Gierens, K., Penner, J., Petzold, A., Prather, M., Schumann, U., Bais, A., Berntsen, T., Iachetti, D., Lim, L., and Sausen, R.: Transport impacts on atmosphere and climate: Aviation, Atmospheric Environment, 44, 4678–4734, https://doi.org/https://doi.org/10.1016/j.atmosenv.2009.06.005, 2010.

Lee, D., Fahey, D., Skowron, A., Allen, M., Burkhardt, U., Chen, Q., Doherty, S., Freeman, S., Forster, P., Fuglestvedt, J., Gettelman, A., De León, R., Lim, L., Lund, M., Millar, R., Owen, B., Penner, J., Pitari, G., Prather, M., Sausen, R., and Wilcox, L.: The contribution of global aviation to anthropogenic climate forcing for 2000 to 2018, Atmospheric Environment, 244, 117 834, https://doi.org/10.1016/j.atmosenv.2020.117834, 2021.

Megill, L., Deck, K., and Grewe, V.: Alternative Climate Metrics to the Global Warming Potential Are More Suitable for Assessing Aviation Non-CO2 Effects, Communications Earth & Environment, 5, 249, 2024.

Prather, M. J., Gettelman, A., and Penner, J. E.: Trade-offs in aviation impacts on climate favour non-$CO_2$ mitigation, Nature, 643, 988–993, https://doi.org/10.1038/s41586-025-09198-2, 2025.

Thor, R. N., Niklaß, M., Dahlmann, K., Linke, F., Grewe, V., and Matthes, S.: The $CO_2$ and non-$CO_2$ climate effects of individual flights: simplified estimation of $CO_2$ equivalent emission factors, Geoscientific Model Development Discussions, 2023, 1–24, https://doi.org/10.5194/gmd-2023-126, 2023.